# Nasal Bacteriomes of Patients with Asthma and Allergic Rhinitis Show Unique Composition, Structure, Function and Interactions

**DOI:** 10.3390/microorganisms11030683

**Published:** 2023-03-07

**Authors:** Marcos Pérez-Losada, Eduardo Castro-Nallar, José Laerte Boechat, Luis Delgado, Tiago Azenha Rama, Valentín Berrios-Farías, Manuela Oliveira

**Affiliations:** 1Computational Biology Institute, Department of Biostatistics & Bioinformatics, Milken Institute School of Public Health, The George Washington University, Washington, DC 20052, USA; 2CIBIO-InBIO, Centro de Investigação em Biodiversidade e Recursos Genéticos, Universidade do Porto, Campus Agrário de Vairão, 4485-661 Vairão, Portugal; 3Departamento de Microbiología, Facultad de Ciencias de la Salud, Campus Talca, Universidad de Talca, Avda. Lircay s/n, Talca 3460000, Chile; 4Centro de Ecología Integrativa, Campus Talca, Universidad de Talca, Avda. Lircay s/n, Talca 3460000, Chile; 5Serviço de Imunologia Básica e Clínica, Departamento de Patologia, Faculdade de Medicina, Universidade do Porto, 4200-319 Porto, Portugal; 6Centro de Investigação em Tecnologias e Serviços de Saúde (CINTESIS@RISE), Faculdade de Medicina, Universidade do Porto, 4200-319 Porto, Portugal; 7Serviço de Imunoalergologia, Centro Hospitalar Universitário São João (CHUSJ), 4200-319 Porto, Portugal; 8i3S—Instituto de Investigação e Inovação em Saúde, Universidade do Porto, 4200-135 Porto, Portugal; 9Ipatimup—Instituto de Patologia e Imunologia Molecular da Universidade do Porto, 4200-135 Porto, Portugal

**Keywords:** 16S rRNA, allergy, asthma, bacteriome, nasal microbiome, rhinitis

## Abstract

Allergic rhinitis and asthma are major public health concerns and economic burdens worldwide. However, little is known about nasal bacteriome dysbiosis during allergic rhinitis, alone or associated with asthma comorbidity. To address this knowledge gap we applied 16S rRNA high-throughput sequencing to 347 nasal samples from participants with asthma (AS = 12), allergic rhinitis (AR = 53), allergic rhinitis with asthma (ARAS = 183) and healthy controls (CT = 99). One to three of the most abundant phyla, and five to seven of the dominant genera differed significantly (*p* < 0.021) between AS, AR or ARAS and CT groups. All alpha-diversity indices of microbial richness and evenness changed significantly (*p* < 0.01) between AR or ARAS and CT, while all beta-diversity indices of microbial structure differed significantly (*p* < 0.011) between each of the respiratory disease groups and controls. Bacteriomes of rhinitic and healthy participants showed 72 differentially expressed (*p* < 0.05) metabolic pathways each related mainly to degradation and biosynthesis processes. A network analysis of the AR and ARAS bacteriomes depicted more complex webs of interactions among their members than among those of healthy controls. This study demonstrates that the nose harbors distinct bacteriotas during health and respiratory disease and identifies potential taxonomic and functional biomarkers for diagnostics and therapeutics in asthma and rhinitis.

## 1. Introduction

Asthma is a chronic inflammatory disorder of the airways induced by complex interactions between the environment and the individual’s genetic and clinical background [1,2]. The onset of asthma results in airway inflammation and mucous production with bronchial obstruction and hyperresponsiveness [3,4,5]. Asthma is a global economic burden with high direct and indirect medical costs [6,7]. It affects people of all ages, being the most common chronic disease among children [8,9]. Over 300 million patients worldwide have been diagnosed with asthma, corresponding to more than 495,000 deaths per year [3,8,10,11]. In Portugal there are almost 695,000 individuals with asthma, corresponding to a prevalence of 8.4% in children and adolescents and 6.8% in adults [12,13,14].

Allergic rhinitis is also a common chronic airway disease worldwide with a substantial economic impact mainly attributable to prescription medications [7,15]. Allergic rhinitis refers to nasal symptoms resulting from inflammation or dysfunction of the nasal mucosa caused by an increase in Th2 cytokines that interfere with the nasal epithelial barrier integrity [16,17,18]. Allergic rhinitis is diagnosed by the observations of its typical symptoms (i.e., rhinorrhea, nasal obstruction, sneezing and nasal pruritus) and the demonstration of IgE-mediated sensitization to aeroallergens [19]. Nearly 400 million people suffer from allergic rhinitis worldwide [20]. In Portugal allergic rhinitis has a prevalence of 9–10% in children and adolescents and 26.1% in adults [12,21,22].

Over the last years evidence has been accumulating on the association between asthma and rhinitis [23,24,25,26]. They appear to be interrelated at the epidemiologic and pathophysiologic levels [23,27,28], and when co-existing in the same patient, asthma prevalence and severity are increased by allergic rhinitis [19,29]. In Portugal more than 46% of the patients with asthma also present allergic rhinitis, which is higher than the worldwide estimate of 38% [19,29].

Multiple studies using high-throughput sequencing (HTS), mainly of the 16S rRNA gene, have already demonstrated that the bacterial communities living in the respiratory airways (i.e., airway bacteriome) play a significant role in the onset, development and severity of both asthma [30,31,32,33,34,35,36,37,38,39,40,41,42,43,44,45,46] and allergic rhinitis [47,48,49,50,51,52]. Microbial HTS has also shown that the nasal cavity is a major reservoir for opportunistic pathogens (e.g., *Moraxella*, *Streptococcus*, *Haemophilus*, *Neisseria*, and *Staphylococcus*), which can spread to other sections of the respiratory tract and potentially induce asthma, rhinitis and other respiratory illnesses [31,35,36,37,38,39,41,48,49,50,51,53,54,55,56,57,58,59]. The importance of the nasal microbiota as a gatekeeper to respiratory health is well known, and their intimate links to chronic airways disease are beginning to be elucidated [60,61,62]. Several studies (see previous references) have already characterized the nasal microbiome and shown that airway bacteriome composition and structure vary between healthy and asthmatic individuals; less is known, however, about the nasal microbiome of individuals with allergic rhinitis with and without asthma comorbidity [48,52,63]. This is particularly remarkable in some countries like Portugal, where despite the high incidence of these respiratory conditions (see statistics and references above), no study has yet characterized the airway microbiomes of healthy, asthmatic or rhinitic individuals. Hence, whether taxonomic and functional characteristics of the nasal microbiota could contribute to asthma or allergic rhinitis in Portugal remains to be determined. Moreover, defining the relationships between the nasal bacteriomes in healthy and respiratory disease individuals could ultimately improve our understanding of asthma and rhinitis pathophysiology and help identifying broadly applicable prognostic markers [64,65].

In this study we have used 16S rRNA HTS to characterize the nasal bacteriomes of 347 participants from northern Portugal with asthma and allergic rhinitis (with and without comorbid asthma) and healthy controls. We sought to identify distinct bacterial taxonomic and functional profiles (i.e., biological markers) across those four clinical groups and compare their microbial composition, diversity, metabolic functions, and microbe-microbe interactions.

## 2. Materials and Methods

### 2.1. Studied Cohort

ASMAPORT was a cross-sectional study of children and adults designed to find associations between airway microbes and clinical manifestations of asthma and rhinitis. ASMAPORT represents a unique sample of otherwise healthy individuals recruited from northern Portugal attending the outpatient clinic of the Serviço de Imunoalergologia in the Centro Hospitalar Universitário São João from July 2018 to January 2020. Patients suspected to have allergic rhinitis or asthma were enrolled at their first visit and after completing a questionnaire on their clinical history. Individuals showing severe inflammation of the nasal cavity, polyps/mass or nasal crusts, “chronic dry mouth”, periodontal lesions greater than 4 mm, oral abscesses, evidence of precancerous lesions or candidiasis were ineligible. Healthy volunteers from the Porto area with no history of respiratory illness were also enrolled but did not complete the questionary or provided clinical information.

All participants in this study were part of the ASMAPORT Project (PTDC/SAU-INF/27953/2017). This study was approved by the “Comissão de Ética para a Saúde” of the Centro Hospitalar Universitário São João/Faculdade de Medicina (Porto) in March 2017, Parecer_58-17. Written consent was obtained from all independent participants or their legal guardians using the informed consent documents approved by the Comissão de Ética.

The diagnosis of allergic rhinitis was confirmed by an allergy specialist based on clinical criteria (sneezing, rhinorrhea and nasal congestion) and a positive skin prick or specific IgE (ImmunoCAP™ ThermoFisher) test to at least one common inhalant allergen in the region (mites, pollens, molds, cat or dog dander) [66,67]. Diagnosis of asthma was established by the attending physician based in the presence of typical symptoms (wheeze, chest tightness, and cough) in the previous 12 months or a positive bronchodilator responsiveness testing with salbutamol (FEV_1_ reversibility of at least 12% and 200 mL) [68].

### 2.2. Sample Collection

A total of 347 individuals participated in this study (Appendix A). They were distributed into four clinical groups: healthy controls (CT = 99 individuals), asthma (AS = 12), allergic rhinitis (AR = 53), and allergic rhinitis with asthma (ARAS = 183). Samples were collected by swabbing the right and left nostrils. We tilted the patient’s head back 70 degrees, inserted the swab less than one inch into the nostril and rotated several times against the nasal wall for about 30 s. We then repeated the process in the other nostril using the same swab. Sample swabs were then preserved in tubes containing DNA/RNA Shield (Zymo Research) and stored at −20 °C until further analysis. Because of the sample size of the AS group, we have only used AS in some of the pairwise comparisons and applied statistical tests that are moderately robust to small sample sizes (see below). Similar considerations were also implemented in other microbiome studies of asthma and rhinitis including groups of ≤12 participants [34,39,50,55,69].

### 2.3. 16S rRNA High-Throughput Sequencing

Total DNA was extracted from swabs using the ZymoBIOMICS™ DNA Miniprep Kit D4300. All extractions yielded <2 ng/μL of total DNA, as indicated by NanoDrop 2000 UV-Vis Spectrophotometer measuring. DNA extractions were prepared for sequencing using the Schloss’ MiSeq_WetLab_SOP protocol in Kozich et al. [70]. Each DNA sample was amplified for the V4 region (~250 bp) of the 16S rRNA gene and libraries were sequenced in a single run of the Illumina MiSeq sequencing platform at the University of Michigan Medical School. Negative controls processed as above showed no PCR band on an agarose gel. We used 10 water and reagent negative controls and 7 mock communities (i.e., reference samples with a known composition) to detect contaminating microbial DNA within reagents and measure the sequencing error rate. We did not find evidence of contamination and our sequencing error rate was as low as 0.0062%.

### 2.4. Microbiome Analyses

16S rRNA–V4 amplicon sequence variants (ASV) in each sample were inferred using dada2 version 1.18 [71]. Exact sequence variants provide a more accurate and reproducible description of amplicon-sequenced communities than is possible with operational taxonomic units defined at a constant level (97% or other) of sequence similarity [71]. Reads were filtered using standard parameters, with no uncalled bases, maximum of 2 expected errors and truncating reads at a quality score of 2 or less. Forward and reverse reads were truncated after 150 bases, merged and chimeras were identified. Taxonomic assignment was performed against the Silva v138.1 reference database using the implementation of the RDP naive Bayesian classifier available in the dada2 R package [72,73]. ASV sequences (226 to 260 bp) were aligned in MAFFT [74] and used to build a tree with FastTree [75]. The resulting ASV tables and phylogenetic tree were imported into phyloseq [76] for further analysis. Sequence files and associated metadata and BioSample attributes for all samples used in this study have been deposited in the NCBI (PRJNA913468). Metadata and ASV abundances with corresponding taxonomic classifications are presented in Appendix A, respectively.

We normalized our samples using the negative binomial distribution as recommended by McMurdie and Holmes [77] and implemented in the Bioconductor package DESeq2 [78]. This approach simultaneously accounts for library size differences and biological variability and has increased sensitivity if groups include less than 20 samples [79]. Taxonomic and phylogenetic alpha-diversity (within sample) were estimated using Chao1 richness and Shannon, ACE, and Phylogenetic (Faith’s) diversity indices. Beta-diversity (between-sample) was estimated using phylogenetic Unifrac (unweighted and weighted), Bray–Curtis and Jaccard distances, and dissimilarity between samples was explored using principal coordinates analysis (PCoA).

Differences in taxonomic composition (phyla and genera) and alpha-diversity indices between respiratory disease groups (AS, AR and ARAS) and healthy individuals (CT) were assessed using the Wilcoxon and the Kruskal–Wallis rank sum tests and the Wald test with Cook’s distance correction for outliers (DESeq2 package), while accounting for covariables (age, season and sex). Beta-diversity indices were compared using permutational multivariate analysis of variance (adonis) as implemented in the vegan R package [80], while also accounting for covariables. None of the covariables were significant for any of the taxonomic and diversity indices compared. We applied the Benjamini–Hochberg method at alpha = 0.05 to correct for multiple hypotheses testing [81,82]. All the analyses were performed in R [83] and RStudio [84]. A full record of all statistical analyses was created in R studio and is included in Appendix A. All data files and R code used in this study with instructions can be found here GitHub (https://github.com/mlosada323/asmaport_bacteriome_nasal, accessed on 5 January 2023).

### 2.5. Functional Analyses

The metagenome functional component of the nasal bacteriome was predicted by coupling Phylogenetic Investigation of Communities by Reconstruction of Unobserved States (PICRUSt2) [85] and the Integrated Microbial Genomes & Microbiomes (IMG/M) database [86]. The hsp.py script was executed with default parameters (maximum parsimony). ASVs abundances were normalized by 16S rRNA gene copy number. Gene abundances were estimated by multiplying the normalized ASV counts by the predicted gene copy numbers using the metagenome_pipeline.py script. From gene abundance predictions, metabolic pathways were predicted using the MetaCyc database [87,88] and the PICRUSt2 pathway_pipeline.py script with default parameters. Differentially abundant metabolic pathways along the different clinical groups were analyzed using the DESEq2 Wald’s test with a *p*-value cutoff of 0.05 and an absolute fold change acceptance criterion of two units.

### 2.6. Network Analyses

To gain insight into community interactions among bacterial taxa in the nasal bacteriome, we generated microbial association networks using the SPIEC-EASI (SParse InversE Covariance Estimation for Ecological Association Inference) R package [89]. All ASVs were classified to their best-hit taxonomic assignment and then agglomerated by identical taxonomic rank. The most parsimonious network structures were detected using LASSO regularized regression by calling the neighborhood selection method (method = “mb”) on the inverse covariance matrix. In order to capture the optimal network links, optimal lambda values were chosen by nlambda = 50 and lambda.min.ratio = 0.01 using 50 subsamples for graph re-estimation (rep.num = 50). Count data were centered log-ratio transformed. The number of links per node was chosen as the centrality metric to detect hub nodes (Degree centrality metric) and the clustering/modulation stage was performed with the default method for the association matrices (cluster_fast_greedy) using the NetCoMi R package [90]. All nodes whose normalized degree centrality metric was greater than the 90 percentile were defined as hub nodes. Finally, network visualizations were generated using the NetCoMi plot function.

## 3. Results

We collected nasal swabs from a cohort of 347 participants (248 individuals with respiratory disease and 99 healthy controls) from northern Portugal comprised mainly of children and young adults (Appendix A). The median age of the participants was 12.6 ± 5.2 years and 52.7% were female. Subjects with respiratory disease were subdivided into three groups: AS (12 subjects), AR (53) and ARAS (183). We sequenced the variable V4 region of the 16S rRNA gene to characterize the nasal bacteriome of each participant. ASV singletons and two CT samples and one ARAS sample with < 1771 reads were eliminated.

### 3.1. Bacteriome Taxonomic Diversity and Structure

The nasal microbiome (344 samples after quality control) comprised 6,515,609 clean reads, ranging from 1771 to 82,430 sequences per sample (mean = 18,940.7) and comprising 6195 ASVs (Appendix A). CT samples had 651 unique ASVs, AS samples had 181, AR samples had 927 and ARAS samples had 2987 (Appendix A). The four groups shared 268 ASVs, while other pairs and trios shared a variable number, ranging from 1 to 363 ASVs (Appendix A).

The nasal bacteriome sequences across all 344 filtered samples were classified into four dominant (<2% abundance) Phyla: Firmicutes (44.9%), Actinobacteriota (27.7%), Proteobacteria (20.3%) and Bacteroidota (4.6%) (Figure 1). Those Phyla comprised 10 dominant (<2%) genera: *Corynebacterium* (21.9%), *Staphylococcus* (18.3%), *Dolosigranulum* (10.6%), *Moraxella* (8.8%), *Streptococcus* (5.2%), *Lawsonella* (3.9%), *Anaerococcus* (2.8%), *Haemophilus* (2.8%), *Neisseriaceae* sp. (2.7%) and *Peptoniphilus* (2.4%) (Figure 1). All the other detected phyla and genera accounted for <2% of the total 16S rRNA sequences each.

Two ASVs (ASV1 and ASV2) of the species *Streptococcus oralis* and *Staphylococcus aureus* comprised the nasal core microbiome (prevalence < 90%) and accounted for 3.5% and 17.1% of the total reads, respectively. The same two ASVs and species comprised the nasal core microbiome of respiratory disease patients and accounted for 4.0% and 17.4% of their total reads, respectively; while only *Staphylococcus aureus* (ASV2) composed the nasal core microbiome of healthy individuals and accounted for 16.6% of the reads. These two core ASVs may represent the more stable and consistent members of the nasal bacteriomes [91,92].

We also compared the mean relative abundance of specific taxa in subjects with respiratory disease and healthy controls. Of the four dominant bacterial phyla comprising the nasal microbiome (Figure 1), one to three phyla showed significant differences in their mean relative proportions between a respiratory disease group (AS, AR or ARAS) and healthy controls (CT), while only Firmicutes varied significantly between AR and ARAS (Table 1). Similarly, of the 10 dominant bacterial genera comprising the nasal microbiome (Figure 1), 5 to 7 genera showed significant differences in their mean relative proportions between a respiratory disease group (AS, AR or ARAS) and CT. However, only two genera (*Anaerococcus* and *Staphylococcus*) varied significantly between AR and ARAS (Table 1). All these significant associations (Wilcoxon test) between phyla and genera and clinical groups were confirmed by the Wald test with Cook’ s distance correction for outliers (0.02 ≤ *p* ≤ 0.0001).

Alpha-diversity indices (Shannon, Chao1, ACE, and PD) of microbial community richness and evenness varied among clinical groups (Figure 2 and Appendix A). AR showed the highest diversity for all indices, while CT showed the lowest. ARAS–CT and AR–CT comparisons were significantly distinct for the four indices (Wilcoxon test; *p* ≤ 0.0026). All the other pairwise comparisons were not significant.

To characterize the structure of the nasal bacteriomes (beta diversity), we applied principal coordinates analysis (PCoAs) to Unifrac (unweighted and weighted), Bray–Curtis and Jaccard distance matrices. All the PCoAs showed partial segregation of the bacteriotas from each clinical group (Figure 3). Subsequently, the adonis analyses detected significant differences (*p* < 0.011) in beta-diversity between each of the respiratory disease groups (AS, AR and ARAS) and the healthy controls for all the distances. None of the pairwise comparisons between respiratory disease groups resulted significant. This suggests that the bacteriomes of AS, AR and ARAS participants may differ from those of healthy individuals in a similar compositional manner.

### 3.2. Bacteriome Functional Diversity

We predicted bacterial functional profiles for the AR, ARAS and CT groups in the nasal mucosa (Appendix A)—the AS group was excluded due to its inadequate sample size for this analysis. We then compared AR and ARAS against control subjects and inferred differentially abundant pathways with *p* < 0.05 and log2FC < 2 (Appendix A). We detected 72 (55 upregulated and 17 downregulated) pathways in AR vs. CT and 72 (50 upregulated and 22 downregulated) pathways in ARAS vs. CT, but only 18 (2 upregulated and 16 downregulated) pathways in ARAS vs. AR. The first two comparisons shared 49 upregulated and 16 downregulated pathways out of 72 differentially expressed pathways; this, again, may suggest that bacteriomes of AR and ARAS participants deviate from those of healthy individuals in a similar manner. Most of those pathways were related to degradation (32–33 pathways) and biosynthesis (23–24 pathways) processes. The AR vs. ARAS comparison was dominated by degradation (9 pathways), fermentation (4 pathways), and biosynthesis (3 pathways) processes.

### 3.3. Bacteriome Network Interactions

We inferred potential interactions among nasal bacteria in the AR, ARAS, and CT groups—AS was again excluded due to its limited sample size. The inferred co-occurrence SPIEC-EASI networks included the following parameters: modules (subnetworks), nodes and hub nodes (key taxa), and connected nodes (Figure 4). The nasal microbial networks of respiratory disease groups (AR and ARAS) were more complex than that of the control group, in accordance with observed trends in intra-group diversity (Figure 2; Appendix A). The CT network included six modules, three hub taxa (*Neisseria*, *Leptotrichia* and *Novosphingobium*), 53 nodes, and 39 connected nodes. The AR network included 17 modules, six hub taxa (*Leptotrichia*, *Novosphingobium*, *Ezakiella*, *Veillonela*, *Actinomyces* and *Corynobacterium* 1 *kroppenstedtii*), 119 nodes, and 92 connected nodes. This network shared two hub taxa with the CT network (*Leptotrichia* and *Novosphingobium*) and also included a subnetwork between *Moraxella* and *Dolosigranulum pigrum*, two taxa usually associated with inflammation [45,50,93]. The ARAS network included 12 modules, nine hub taxa (*Aliterella_CENA595*, *Deinococcus*, *Leptotrichia*, *Neisseria*, *Veillonela*, *Gemella*, *Actinomyces*, *Finegoldia magna* and *Johnsonella*), 109 nodes, and 68 connected nodes. Two hub taxa were also shared with the CT network (*Leptotrichia* and *Neisseria*) and three with the AR network (*Leptotrichia*, *Veillonela* and *Actinomyces*). Of the 10 dominant genera in the nasal bacteriome (Table 1), 6 formed subnetworks in CT, 8 in AR and 10 in ARAS. Interestingly, *Moraxella* and *Staphylococcus* only appeared in the networks of rhinitic patients alone or connected to the opportunistic pathogen *Dolosigranulum pigrum* [94]. Similarly, other commensal genera (e.g., *Corynebacterium* and *Veillonella*) were also associated in separate modules, suggesting a robust relationship.

## 4. Discussion

Asthma and allergic rhinitis are two conditions that, either when occurring together or separately, impart a health and economic burden to persons and society [6,7,15,95,96,97]. Emerging evidence has suggested that both airway diseases are intimately linked to alterations of the nasal bacteriome [45,52,56,93,98,99,100,101,102]. In this cross-sectional study, we apply 16S rRNA amplicon HTS to a large cohort of individuals from northern Portugal with asthma or allergic rhinitis (with and without comorbid asthma) and healthy controls to characterize their nasal bacteriotas. We identified distinct taxonomic and functional bacterial profiles and co-occurrence networks associated with chronic respiratory disease.

The nasal bacteriomes of the studied samples were composed of four dominant phyla and 10 dominant genera (Figure 1 and Table 1). All these taxa have been previously described in the nasal cavity of asthmatic, rhinitic or healthy individuals [35,37,38,47,48,49,51,52,69,103,104], where they are considered normal residents. The characterized bacteriotas were mainly comprised of commensal taxa [103,104], but some genera (e.g., *Moraxella*, *Streptococcus*, *Haemophilus*, *Neisseria* and *Staphylococcus*) including pathogenic species associated to asthma [33,34,35,36,39,40,55,57,58,59,105] and allergic rhinitis [47,48,49,50,51,52] were also detected. Hence, overall, the nasal bacteriome of children and young adults from northern Portugal resembled those described in other studies of cohorts from USA, Europe, Australia and Asia.

Both heathy participants and those with a chronic respiratory disease harbored unique microbial taxa in their nasal mucosa. The healthy nasal bacteriome contained 10.5% unique ASVs, while the AS, AR and ARAS bacteriomes contained 2.9%, 15% and 48.2% unique ASVs, respectively (Appendix A). These ASVs may represent fingerprints or biomarkers in those patients with asthma and allergic rhinitis with and without comorbid asthma. Future microbiome studies will need to confirm their consistency across other cohorts and nasal microenvironments [41,106], and their potential as targets for new therapeutic strategies in asthma and rhinitis [34,39,107].

The proportions of most of the dominant bacterial phyla and genera in the nose varied significantly between healthy and respiratory disease groups (Table 1). The most significant differences in phyla (three out of four) and genera (7 out of 10) and relative mean abundance (Figure 1) were observed in CT vs. AR and CT vs. ARAS. Nonetheless, one phylum and five genera varied significantly between AS and CT, despite the small sample size of the AS group. Actinobacteriota was more abundant in CT, while Firmicutes, Proteobacteria and Bacteroidota were more abundant in the disease groups. Similarly, *Corynebacterium* and *Lawsonella* were more abundant in healthy subjects, while *Dolosigranulum, Haemophilus, Moraxella* and *Streptococcus* were more abundant in all the respiratory disease groups. Firmicutes, *Anaerococcus* and *Staphylococcus* increased significantly in rhinitic participants with comorbid asthma compared to those without (Table 1). A similar study in Chinese adult participants showed the opposite trend for the phylum, but the same result for *Staphylococcus* [48]. The compositional patterns observed here agree well with some previous studies of asthma and allergic rhinitis [48,49,69,108,109], and confirm the pathogenic potential of some of these genera (e.g., *Haemophilus, Moraxella* and *Streptococcus*) via host inflammatory or immune response [34,39,45,109]. Changes in these bacterial groups may then provide insight into the pathobiology of asthma and allergic rhinitis. Nonetheless, given the diversity (asthma) and limitation (allergic rhinitis) of microbiome studies so far, intrasubject variation, lack of biological and longitudinal replicates, and limited resolution of 16S rRNA HTS, the relationships between specific bacterial colonization, dysbiosis and chronic inflammatory disease may still remain elusive [50,93].

Bacterial alpha-diversity (species richness and evenness) varied significantly between samples from healthy controls and those from participants with allergic rhinitis, with and without comorbid asthma (Figure 2). No differences were observed for the four indices between AR and ARAS. Alpha-diversity has shown inconsistent patterns in the upper airway bacteriome. Some studies have revealed less within-sample diversity in healthy controls compared to asthma [69,110,111] or allergic rhinitis with and without comorbid asthma [49,51,112]; while others have shown the opposite trend across those same groups [48,50,93,109,113] or across metrics of richness and evenness [34,109]. A recent study has suggested that asthma may substantially affect alpha-diversity more than AR in the upper airway, since AR values are not as low as AS values compared to CT [48]. Our results seem to confirm that statement (Figure 2; Appendix A). Nonetheless, given the discrepancy in alpha-diversity patterns in microbial studies of asthma and allergic rhinitis, this metric may not be a good proxy of disease status or pathogenesis in the nose.

All nasal bacterial communities in samples from respiratory disease participants (AS, AR and ARAS) were significantly restructured compared to those from healthy controls (Figure 3). No differences were observed, however, between AR and ARAS groups. This pattern held irrespective of the distance metric used, whether accounting for phylogenetic diversity or not. Previous studies also showed specific community structuring associated with distinct bacterial composition among these same groups [48]; while others also confirmed structural differences between healthy controls and patients with asthma [34,69,109,113] or allergic rhinitis [47,50,51]—although one study reported conflicting results for the latter [49]. It is well established that altered bacterial diversity increases the risk of immune-mediated diseases like asthma and allergic rhinitis [35,45,114], but while alpha-diversity might not be a consistent predictor of disease status in the nasal microbiome, beta-diversity indices may be more reliable indicators of heterogeneity/stochasticity associated to dysbiosis [115,116]. As reported for the human gut microbiome, we speculate that the human airway bacteriome may also follow the Anna Karenina principle, i.e., “all healthy microbiomes are alike, but each disease-associated microbiome (i.e., asthma, allergic rhinitis and their combined occurrence) is sick in its own way”.

The airway microbiota can influence host metabolism and homeostasis, including epithelial cell growth and repair, and inflammatory and immune responses, thereby impacting chronic disease onset and progression [34,39,45,99,100,101,102,109,117,118,119]. Compared to healthy controls, our PICRUSt2 analyses predicted 50–55 pathways upregulated in rhinitic patients (Appendix A). A similar array of differentially expressed pathways was also observed in a previous study comparing AR to CT participants [109]. As far as we know, that and ours are the only two studies so far using PICRUSt2 to predict metabolic functions in the nasal microbiome during allergic rhinitis. Several of the inferred metabolic pathways (e.g., tryptophan, tyrosine, histidine, nicotinate, acetate or glycerol metabolism) have been associated with allergic sensitization and inflammation of the airways [39,120,121,122,123]. Our study, thus, suggests that dysbiosis of the nasal bacteriome may influence these bacterial metabolic pathways, thereby affecting the development of allergic rhinitis. Nonetheless, since microbial function here has been predicted using 16S rRNA amplicons, more powerful dual-transcriptomic studies, e.g., [34,39], should be performed to confirm our predictions and decipher the interplay between host and microbiota.

Finally, co-occurrence network analyses revealed distinct and specific connectivity patterns (i.e., interactions) in rhinitic groups compared to healthy controls (Figure 4). The AR and ARAS networks were more intricate than that of the control group including more and larger modules and connected taxa. Different modules in each network represent different co-regulated bacteria that, in turn, suggest distinct community partitions [124]. Many microbes of variable abundance were embedded in the networks, highlighting their importance individually and also in the community (interactions). Highly connected taxa (e.g., *Veillonella* and *Leptotrichia*), even if in low abundance, may still play key roles in the functionality of the nasal community [124,125]. Microbes that are both prevalent and abundant (e.g., core taxa *Streptococcus* and *Staphylococcus*) in the nose of patients with AR and ARAS, but are also highly connected, might serve as better indicators of disease [126,127,128]. Similarly, understudied bacteria in AR and ARAS patients connected to well-known pathogenic groups may also be drivers of disease [129]. Microbe–microbe interactions have not been investigated in rhinitis. However, a few studies in asthma have also revealed striking differences between networks of asthmatic groups and healthy controls, although with opposite trends in connecting density [38,124,125,129]. Further research is still needed to assess the role of microbial networks and their biomarker potential in the pathogenesis of inflammation [98].

The bacteriomes of patients with AR and ARAS showed some differences in composition for specific taxa (see comments above and Table 1), but no differences in alpha- and beta-diversity were observed at the community level. A previous study [48] described differences in intra-and inter-sample diversity between AR and ARAS bacteriomes, although all the enrolled participants were adults (mean age/group < 37 years). Despite few significant changes in composition, we observed some significant variation in the functionality of AR and ARAS (Appendix A) and a richer pattern of microbe–microbe interactions in AR (Figure 4), but not as remarkable as the difference observed between rhinitic patients and controls. Therefore, by comparison, nasal bacterial communities across respiratory disease groups varied much less, which may suggest that, at least in our cohort, the etiology and pathophysiology of these two chronic respiratory illnesses may be driven by a shared group of bacteria.

## 5. Conclusions

We characterized for the first time in Portugal the nasal bacteriomes of individuals with asthma and rhinitis (with and without comorbid asthma) and healthy controls. We demonstrated that several of the most abundant bacterial phyla and genera in the nose varied significantly between healthy and respiratory disease participants (i.e., potential biomarkers of disease). We also showed that their nasal bacteriotas are compositionally and structurally distinct, encode different metabolic functions and establish different microbe-microbe connections among their members. This study, hence, confirms that bacterial diversity, function and interactions contribute to the pathogenesis of asthma and allergic rhinitis [45,56,93,98,99,100,101,102], and generates new insights into the relationship between nasal bacteriome and airway mucosal inflammation.

## Figures and Tables

**Figure 1 microorganisms-11-00683-f001:**
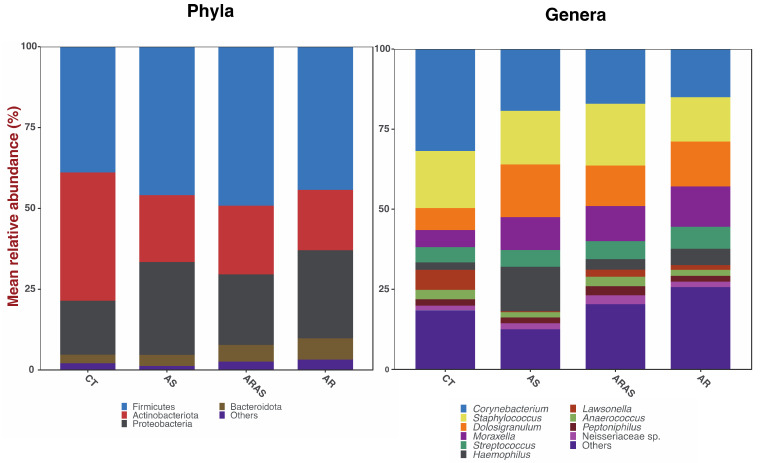
Bar plots of mean relative proportions of the top bacterial phyla and genera in the nasal bacteriome of participants with asthma (AS), allergic rhinitis with comorbid asthma (ARAS), allergic rhinitis (AR) and healthy controls (CT).

**Figure 2 microorganisms-11-00683-f002:**
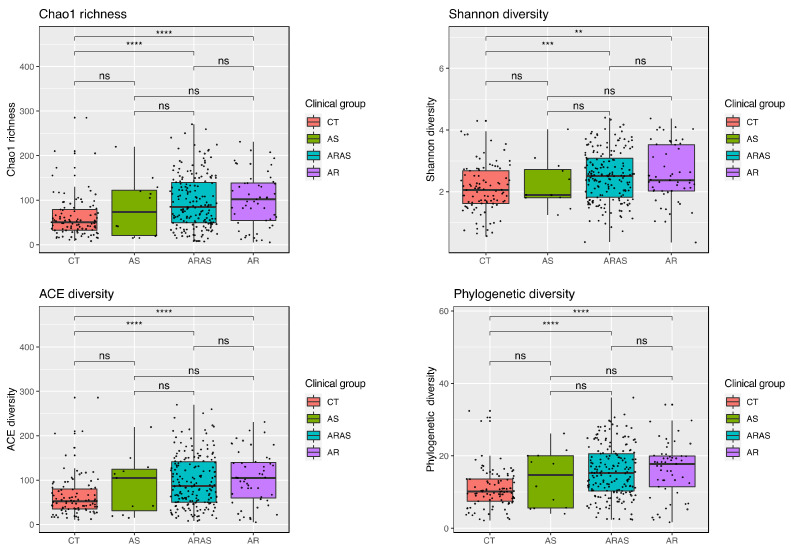
Alpha-diversity estimates (Chao1, Shannon, ACE, and phylogenetic diversity) of nasal bacterial diversity and statistical significance (Wilcoxon test) in participants with asthma (AS), allergic rhinitis with comorbid asthma (ARAS), allergic rhinitis (AR) and healthy controls (CT). ns = not significant; ** = *p* ≤ 0.01; *** = *p* ≤ 0.001; **** = *p* ≤ 0.0001.

**Figure 3 microorganisms-11-00683-f003:**
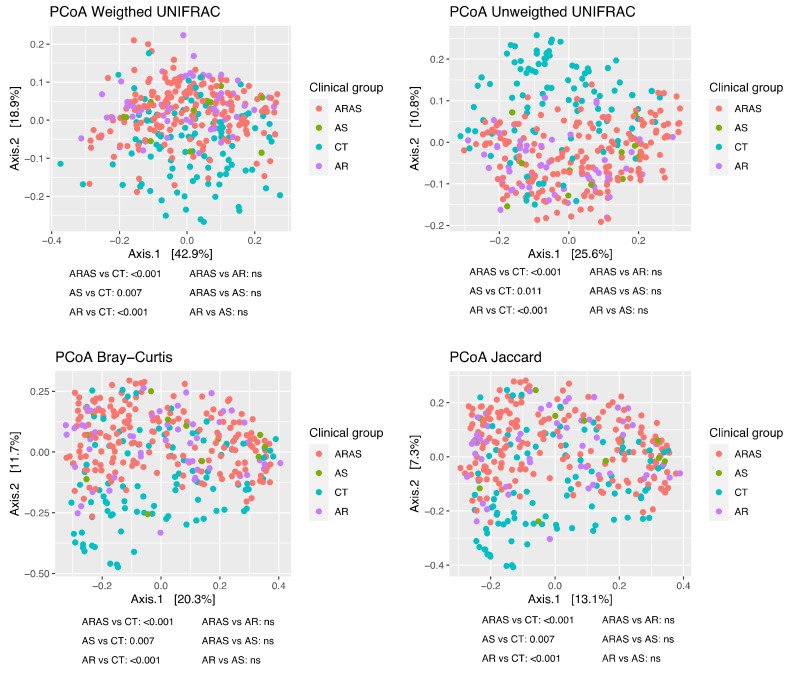
Principal coordinates analysis (PCoA) plots of beta-diversity estimates (Unifrac, Bray–Curtis and Jaccard indices) and statistical significance (adonis test) of the nasal bacteriome of participants with asthma (AS), allergic rhinitis with comorbid asthma (ARAS), allergic rhinitis (AR) and healthy controls (CT). ns = not significant.

**Figure 4 microorganisms-11-00683-f004:**
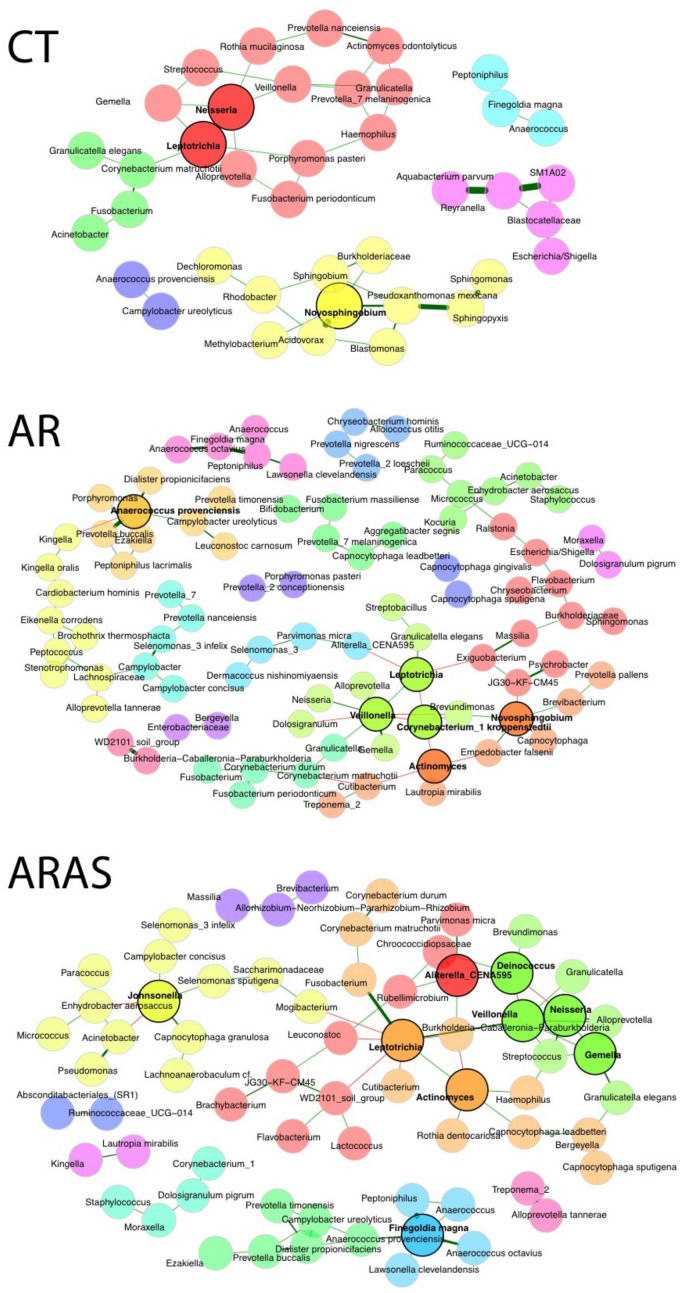
Co-occurrence networks of bacterial taxa in the nasal bacteriome of participants with rhinitis with comorbid asthma (ARAS), allergic rhinitis (AR) and healthy controls (CT). Nodes represent taxa connected by edges whose width is proportional to the strength of their association. Green and red edges indicate positive and negative correlations, respectively. Nodes within the 90th percentile of degree connectivity are considered hub nodes (black circle line). Inferred modules or subnetworks are colored differently.

**Table 1 microorganisms-11-00683-t001:** Mean relative proportions and statistical significance of pairwise comparisons (Wilcoxon test) of bacterial genera and phyla in the nasal microbiome of participants with asthma (AS), allergic rhinitis with comorbid asthma (ARAS), allergic rhinitis (AR) and healthy controls (CT). ns = not significant.

	Mean Relative Proportions (%)					Wilcoxon Test Significance				
	CT	AS	ARAS	AR	AS-AR	AS-ARAS	ARAS-AR	AS-CT	ARAS-CT	AR-CT
Phylum										
Actinobacteriota	42.0	21.1	21.1	18.1	ns	ns	ns	0.021	<0.001	<0.001
Bacteroidota	2.3	4.1	5.3	6.5	ns	ns	ns	ns	<0.001	<0.001
Firmicutes	38.4	49.8	50.4	43.7	ns	ns	0.036	ns	<0.001	ns
Proteobacteria	15.4	23.6	20.6	28.7	ns	ns	ns	ns	ns	<0.001
Others	1.9	1.4	2.5	3.1	-	-	-	-	-	-
Genus										
*Anaerococcus*	2.9	2.2	3.1	2.1	ns	ns	0.031	ns	ns	ns
*Corynebacterium*	33.2	19.3	16.6	14.4	ns	ns	ns	ns	<0.001	<0.001
*Dolosigranulum*	7.5	15.3	12.2	13.0	ns	ns	ns	0.021	<0.001	<0.001
*Haemophilus*	1.6	10.0	2.7	4.2	ns	ns	ns	0.002	<0.001	<0.001
*Lawsonella*	7.3	0.5	2.5	1.4	ns	ns	ns	0.001	<0.001	<0.001
*Moraxella*	5.2	8.6	9.3	15.3	ns	ns	ns	0.002	<0.001	<0.001
*Neisseriaceae* sp.	2.8	2.1	3.9	2.3	ns	ns	ns	0.018	<0.001	<0.001
*Peptoniphilus*	1.9	2.7	2.8	1.9	ns	ns	ns	ns	ns	ns
*Staphylococcus*	18.0	19.0	20.4	13.0	ns	ns	0.013	ns	ns	ns
*Streptococcus*	3.8	6.0	5.9	7.2	ns	ns	ns	ns	<0.001	<0.001
Others	15.8	14.4	20.8	25.2	-	-	-	-	-	-

## Data Availability

Sequence files and associated metadata and BioSample attributes for all samples used in this study have been deposited in the NCBI (PRJNA913468). A full record of all statistical analysis is included as Appendix A and was created in R studio. Metadata and ASV table with corresponding taxonomic classifications have all been included as Appendix A, respectively. Data files, R code and instructions to execute all the analyses are available in GitHub (https://github.com/mlosada323/asmaport_bacteriome_nasal, accessed on 5 January 2023).

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
