# Peer review of "Nasal Bacteriomes of Patients with Asthma and Allergic Rhinitis Show Unique Composition, Structure, Function and Interactions"

_microorganisms, 2023, doi:10.3390/microorganisms11030683_

Round 1

Reviewer 1 Report

This is a well done research dealing with a Novel focus in rhinology literature. 
i would like to know more about the nasal sample collections. It would be explained better.

Author Response

Thanks for your kind evaluation. We've added the following information regarding nasal sample collections. "We tilted patient's head back 70 degrees, inserted the swab less than one inch into the nostril and rotated several times against the nasal wall for about 30 seconds. We then repeated the process in the other nostril using the same swab.". 

Reviewer 2 Report

 This is a very interesting and well writen study. As an immunologist , in my opinion the results support in the "one airway concept".

I find a bit of difficulty understand the methods regarding the microbiome analysis. Maybe the authors should consider to explain these methods with simplisity -to physicians like me that are not microbiome analists but will find this study very interseting for them to read. 

Author Response

Thanks for your comment and positive evaluation. We understand the bioinformatic and statistical methods used in microbiome analysis may be hard to understand to the neophyte, so we're providing a detail description of the analyses performed in our supplementary materials (Fig. S1) and GitHub pages. We feel like a very detailed explanation of the methodology is beyond the scope of this article. We've, nonetheless, described our methods in a bit more detail. Incidentally, referee #4 acknowledged the "perfect description of the research methods". I encourage the referee to review the extensive literature and reviews on microbiome research.

Reviewer 3 Report

I appreciate the opportunity to review the manuscript for publication in MDPI Microorganisms.

I feel that the topics are interesting and the manuscript is grossly well organized.

I find the authors' work to be scientifically valuable, well written and of interest.

I have a few comments as follows.

The authors should consider citing the latest article;

Microbiome in Nasal Mucosa of Children and Adolescents with Allergic Rhinitis: A Systematic Review. Children 2023, 10(2), 226; https://doi.org/10.3390/children10020226

L113: “one common inhalant allergen in the region (mites, pollens, molds, cat or dog dander)”

The authors should explain the season when nasal samples were collected for seasonal antigens.

The authors had better disclose possible relation between the severity of the disease and the nasal microbiome features.

Author Response

Thanks for your valuable comments and evaluation. We've added and discussed the review article you suggested [Ref #52]. We have also included seasonal information in supplementary Table S1. Samples were collected throughout the year, all four seasons included. As explained in the methods, we accounted for seasonal variation in the analyses. However, no differences in microbial composition or diversity were detected. In this article we've not explored potential associations between disease severity and microbiome features. We hope, however, to address that and other clinical issues in a future study by our group.

Reviewer 4 Report

An extremely important study on the nasal bacteriome dysbiosis during allergic rhinitis, alone or associated with asthma comorbidity. The authors shouldbe congratulated for their strive: clear definition of the study cohort, perfect description of the research methods - accurate description of the rRNA high-throughput sequencing, of the microbiome analyses and network analyses. The authors clearly demonstrate that that several of the most abundant bacterial phyla and genera in the nose varied significantly between healthy and respiratory disease.The authors demonstrate that bacterial diversity, function and interactions contribute to the pathogenesis of asthma and allergic rhinitis.

Author Response

Thank you very much for your very kind comments.